# Mesenchymal Stem Cell-Derived Exosomes: A Promising Therapeutic Agent for the Treatment of Liver Diseases

**DOI:** 10.3390/ijms231810972

**Published:** 2022-09-19

**Authors:** Yi Ding, Qiulin Luo, Hanyun Que, Nan Wang, Puyang Gong, Jian Gu

**Affiliations:** College of Pharmacy, Southwest Minzu University, No. 16, South 4th Section, First Ring Road, Chengdu 610041, China

**Keywords:** mesenchymal stem cells, exosomes, liver diseases, therapy, regeneration

## Abstract

Liver disease has become a major global health and economic burden due to its broad spectrum of diseases, multiple causes and difficult treatment. Most liver diseases progress to end-stage liver disease, which has a large amount of matrix deposition that makes it difficult for the liver and hepatocytes to regenerate. Liver transplantation is the only treatment for end-stage liver disease, but the shortage of suitable organs, expensive treatment costs and surgical complications greatly reduce patient survival rates. Therefore, there is an urgent need for an effective treatment modality. Cell-free therapy has become a research hotspot in the field of regenerative medicine. Mesenchymal stem cell (MSC)-derived exosomes have regulatory properties and transport functional “cargo” through physiological barriers to target cells to exert communication and regulatory activities. These exosomes also have little tumorigenic risk. MSC-derived exosomes promote hepatocyte proliferation and repair damaged liver tissue by participating in intercellular communication and regulating signal transduction, which supports their promise as a new strategy for the treatment of liver diseases. This paper reviews the physiological functions of exosomes and highlights the physiological changes and alterations in signaling pathways related to MSC-derived exosomes for the treatment of liver diseases in some relevant clinical studies. We also summarize the advantages of exosomes as drug delivery vehicles and discuss the challenges of exosome treatment of liver diseases in the future.

## 1. Introduction

The causes of liver disease are complex, and there are many complications. Therefore, it is a difficult disease to cure. When effective treatments are not administered in a timely manner, the patient’s quality of life and health level are seriously affected [1]. External factors, such as chemical toxicants, viruses and a high-fat diet may cause early liver diseases, which develop into end-stage liver diseases when not detected and treated early [2,3,4]. The degree of liver cell necrosis is high in end-stage liver disease, the necrosis area is large, and the matrix deposition is excessive, which makes it difficult for the liver to regenerate. The main treatment modality for end-stage liver diseases is liver transplantation, but there is a scarcity of suitable donor organs. Transplantation is also an expensive and risky surgery with difficult postoperative recovery. Therefore, the search for a new effective and efficient therapeutics for the treatment of liver disease has been a popular research topic.

Mesenchymal stem cells (MSCs) are adult stem cells with self-renewal and multidirectional differentiation potential that may be obtained from bone marrow, adipose tissue, dental pulp, peripheral blood and other tissues [5]. MSCs have powerful immunomodulatory and tissue repair capabilities, which supports their great potential in regenerative medicine applications [6,7,8]. Experimental studies showed that MSCs were involved in the repair of damaged tissues via homing mechanisms and paracrine and cell–cell contact mechanisms [9,10,11]. However, MSC treatments have some disadvantages. For example, different transplantation methods significantly impact the survival and homing rate of MSCs in vivo. Clinical and preclinical studies showed that different transplantation routes directly affected the biodistribution and survival of MSCs [12,13,14]. The number of in vitro passages affected the homing rate of MSCs in vivo [15], and because MSCs produce a broad spectrum of cytokines, chemokines and growth factors, they may promote tumor growth [16,17]. Therefore, identifying a cell-free therapy to avoid the shortcomings of MSCs in regeneration and repair is worthy of further effort.

More intensive basic experimental and preclinical studies found that MSC-derived exosomes (40–100 nm in diameter) were a possible novel therapeutic tool and provided new insights into the treatment of liver diseases. The therapeutic ability of stem cells primarily occurs via paracrine actions of factors secreted from the stem cells, and exosomes play a major role in paracrine action. MSC-derived exosomes are extracellular vesicles produced by parental cells via the paracrine pathway, and their contents include proteins and nucleic acids, which help maintain tissue homeostasis and restore critical cellular functions by initiating repair and regeneration processes [18,19]. Compared to direct parental cell transplantation, MSC-derived exosomes have low immunogenicity, non-tumorigenicity, easy storage, and high clinical safety. MSC-derived exosomes also mediate a variety of pathological processes, such as inflammation [20], liver diseases [21], kidney diseases [22], cardiovascular diseases [23], and cancer [24], and serve as drug delivery vehicles to effectively protect drug activity with potential therapeutic effects in a variety of diseases [25]. The powerful repair ability of the liver itself may synergize with MSC-derived exosomes to maximize its regeneration ability. Therefore, MSC-derived exosomes may become an ideal therapeutic tool for liver diseases in the near future. The present review presents research progress on MSC-derived exosomes in the treatment of liver diseases to improve the research ideas for the development of a novel cell-free therapy.

We introduce the therapeutic efficacy and molecular mechanism of MSC-derived exosomes in the treatment of liver diseases. We performed an extensive literature review in multiple databases (Google Scholar, Sci-Hub, PubMed.gov, and ClinicalTrials.gov) from 2012 to the present. The main keywords for the selection search were “MSCs”, “extracellular vesicles”, “exosomes”, “liver disease”, “liver injury”, “liver fibrosis”, “liver failure”, “hepatocellular carcinoma”, “drug delivery systems”, and “tumours”. All of the authors reviewed the retrieved literature to ensure relevance to the content of this manuscript and present their own preliminary insights for the manuscript.

## 2. Biogenesis and General Physiological Functions of Exosomes

The process of exosome formation begins with endocytosis, in which the cell membrane and surface proteins invaginate into the internal cellular environment and lead to the formation of early endosomes. Late endosomes develop under the influence of the Golgi apparatus. Late endosomes accumulate large amounts of “cargo” that is sorted into luminal vesicles (ILVs), which is mediated by the endosomal sorting complexes required for the transport (ESCRT)-dependent pathway [26] and the ESCRT-independent pathway [27]. ILVs reaggregate and convert late endosomes into multivesicular bodies (MVBs). One part of an MVB binds to autophagosomes or lysosomes, which disrupts the structure of ILVs and returns the contents to the cytoplasmic matrix. The other part of an MVB fuses with the plasma membrane of the parent cell, and the ILVs are released outside the cell in the form of cytosol, i.e., exosomes (Figure 1).

Exosomes contain a variety of molecules, including proteins, lipids, DNA and RNA. Among the different types of RNA, the presence of microRNAs (miR-) is notable because miRNAs are involved in cell adhesion, membrane fusion, metabolism, and signaling [28,29,30,31,32]. Exosomes contain a variety of proteins, including tetraspanins, integrins (CD9, CD63, CD81 and CD82) involved in cell targeting and adhesion, ALG-2-interacting protein X (Alix) and tumor susceptibility gene 101 protein (TSG101), which are involved in the production of MVBs, and membrane association proteins and floating ship proteins that are important for membrane fusion [33].

Exosomes have a wide range of physiological functions in living organisms, including mediating intercellular communication, substance exchange, the immune response, and cell proliferation and maturation [34,35,36]. Exosomes play important roles in intercellular communication and substance exchange via three main mechanisms: binding to target cell surface receptors; direct fusion with target cell membranes; and entry into target cells via endocytosis [37]. Exosomes stimulate the production of tumor necrosis factor α (TNF-α) and interleukin (IL)-12, and the recruitment of neutrophils and macrophages during the body’s immune response [38]. Exosomes also stimulate the migration of natural killer cells (NK cells) [39] and mediate autoimmune diseases by regulating T cells [40]. Exosomes promote tissue regeneration. Hepatocyte-derived exosomes affect liver repair and regeneration via sphingosine-1-phosphate (S1P), which likely occurs via the direct fusion of exosomes with target cells and the transfer of neutral ceramidase and sphingosine kinase 2 (SK2) to increase S1P synthesis [41]. Most cells are capable of producing exosomes that affect the proliferation and development of the cells themselves and participate in intercellular communication and material exchange, which affect the physiological functions of the organism.

## 3. Contents and Physiological Functions of MSC-Derived Exosomes

Although the biogenesis of MSC-derived exosomes is similar to other sources, their rich and unique contents influence physiological functions. MSC-derived exosomes express the common biomarkers CD9 and CD81 and the MSC-specific adhesion factors CD73, CD44, and CD29 [42]. Proteomic analysis of MSC-derived exosomes from humans identified 730 proteins that affect cell proliferation, adhesion, migration and morphogenesis capacities [43]. Comprehensive proteomic analyses of exosomes derived from human primed MSCs indicated that these exosomes contained a higher proportion of specific protein subclasses and identified abundant fibronectin-mediated cell mitosis, which may provide the molecular basis for their unique functional properties [44].

MSC-derived exosomes are also rich in RNA. The integration of RNA into exosomes is specific, and certain RNAs preferentially accumulate in exosomes [19]. MSC-derived exosomes contain mRNAs associated with differentiation into the mesenchymal phenotype, cell transcription and proliferation [45], but human hepatic stem cell-derived exosomes contain mRNA related to hepatocyte metabolism and proliferation [46]. NanoString, biological process, pathway and pathway target analyses revealed that miRNAs in MSC-derived exosomes were involved in multiple biological processes of vascular development, angiogenesis, cell proliferation and fibrosis, and mediated angiogenic, antifibrotic and antiapoptotic phenotypes [47]. These results suggest that RNA plays important roles in the physiological functions of MSC-derived exosomes.

The enzymes contained in MSC-derived exosomes were also investigated. Proteomic studies showed that MSC-derived exosomes contained five important enzymes involved in glycolysis: glyceraldehyde-3P dehydrogenase, phosphoglycerate kinase (PGK), phosphoglucomutase, enolase, and pyruvate kinase m2 isoform (PKm2). These enzymes catalyze five reactions related to glycolysis and may play a role in adenosine triphosphate (ATP) production [48]. MSC-derived exosomes substantially replenished the deficiency of glycolytic enzymes due to ischemia reperfusion [49]. ATP and nicotinamide adenine dinucleotide (NADH) levels increase, oxidative stress decreases, and local and systemic inflammation is reduced [50].

In conclusion, MSC-derived exosomes have unique surface markers, and the diversity and richness of their contents affect their physiological functions.

## 4. Therapeutic Applications and Mechanisms of MSC-Derived Exosomes in Liver Diseases

MSCs immunomodulate and promote tissue regeneration, and MSC-derived exosomes possess lower immunogenicity and non-tumorigenicity but inherit the properties of parental cells. MSC-derived exosomes are superior to parental stem cells in organ injury treatment regeneration and may become an effective alternative to MSCs for clinical applications. MSC-derived exosomes may cooperate or enhance the self-repair ability of the liver to treat liver diseases. The following sections describe the studies related to MSC-derived exosomes in the treatment of liver diseases (Figure 2).

### 4.1. MSC-Derived Exosomes for the Treatment of Liver Injury

Drugs and chemical poisons cause acute and chronic liver injury. The basic feature of most liver injuries is hepatocyte death, which triggers a large number of inflammatory cell infiltrations [51,52]. Therefore, anti-apoptosis may be a strategy to benefit liver repair. N-acetylcysteine is a U.S. Food and Drug Administration-approved antidote in the treatment of acute liver injury. However, the control of the injection dose and certain liver toxicity are inevitable problems [53,54].

MSC-derived exosomes have cell proliferation-promoting, anti-inflammatory, and M2 macrophage polarization-promoting effects, which produce positive effects in the treatment of liver injury [55]. In vitro D-galactosamine and lipopolysaccharide (D-GalN/LPS)-induced hepatocyte injury and apoptosis models found that BMSC (bone mesenchymal stem cells)-derived exosomes promoted the formation of autophagosomes, significantly decreased the expression levels of the proapoptotic proteins Bax and cleaved caspase-3, and upregulated the expression level of the antiapoptotic protein Bcl-2. These results suggested that BMSC-derived exosomes attenuated D-GalN/LPS-induced hepatocyte injury and apoptosis by inducing autophagy activation [56].

Transplantation of BMSC-derived exosomes into mice with carbon tetrachloride (CCl_4_)-induced liver injury increased hepatocyte proliferation in mice [57]. Human umbilical cord MSC (HUC-MSC)-derived exosomes exerted hepatoprotective effects via antioxidative stress and antiapoptotic cell death in the same experimental model [58]. For immune liver injury triggered by concanavalin-A (Con-A), MSC-derived exosomes improved the area of the hepatic necrotic zone and the degree of apoptosis, and increased the mRNA expression of anti-inflammatory cytokines and regulatory T (Treg) cell numbers, which may be related to the immunomodulatory properties of exosomes [59]. Human-induced pluripotent stem cell-derived mesenchymal stromal cell (hiPSC-MSC)-derived exosomes also ameliorated hepatic ischemia-reperfusion injury (IRI) primarily by activating the SK and S1P signaling pathways, which protects the liver by inhibiting hepatocyte necrosis and hepatic sinusoidal congestion, and increasing proliferating cell nuclear antigen (PCNA) and phospho-histone H3 (PHH3) expression levels [60]. Transplanted human umbilical cord blood MSC (HUCB-MSC)-derived exosomal miR-1246 alleviated hepatic IRI by decreasing the expression of inflammatory factors (such as IL-6, IL-1β, IL-17, and IL-10) and reducing the number of TUNEL-positive cells and the T helper 17 (Th17)/Treg imbalance ratio. Further studies revealed that HUCB-MSC-derived exosomes alleviated hepatic IRI by transporting miR-1246 to regulate the glycogen synthase kinase 3β (GSK3β)-mediated Wnt/β-catenin pathway, which improved hepatic IRI by modulating the balance between Tregs and Th17 cells via the miR-1246-mediated IL-6-gp130-signal transducer and activator of transcription 3 (STAT3) axis [61,62]. IL-6 stimulated the secretion of miR-455-3p-rich exosomes from HUC-MSCs, which inhibited the activation and cytokine production of LPS-activated macrophages in vivo and in vitro, and ameliorated chemical liver injury by reducing macrophage infiltration and local liver injury and lowering serum inflammatory factor levels [63]. These in vivo and in vitro studies show that MSC-derived exosomes attenuate liver injury via anti-inflammatory, anti-apoptotic, anti-oxidant, and hepatocyte proliferation-promoting pathways.

### 4.2. MSC-Derived Exosomes for the Treatment of Liver Fibrosis

Liver fibrosis is the subsequent manifestation of various liver injuries, which are characterized by an excessive and abnormal deposition of extracellular matrix proteins. Activation of hepatic stellate cells (HSCs) is the key link [64]. Unfavorable external factors damage hepatic parenchymal cells, which undergo apoptosis, and inflammatory cells accumulate in the damaged area and release chemokines that activate HSCs. Activated HSCs participate in the formation of liver fibrosis and the reconstruction of intrahepatic structures by proliferating and secreting extracellular matrix proteins and promoting the production of collagen by fibroblasts and bone marrow-derived myeloid fibroblasts, which create the pathological basis of liver fibrosis [65]. Therefore, reducing inflammation, inhibiting HSC proliferation, and reducing collagen deposition improve liver fibrosis.

The ameliorative effect of MSCs on liver fibrosis was demonstrated, and MSC-derived exosomes may exert a similar effect [66]. A recent study showed that MSC-derived exosomal circDIDO1 overexpression inhibited HSC proliferation, reduced profibrotic markers and induced apoptosis and cell cycle arrest. These effects may be mediated by MSC-derived exosome circDIDO1 translocation, which increased phosphatase and tensin homolog deleted on chromosome ten (PTEN) to suppress the protein kinase B (AKT) pathway and inhibit HSC activation by sponging miR-143-3p [67]. Another experiment showed that miR-618 was transferred into LX-2 cells via MSC-derived exosomes and inhibited the viability and migration of LX-2 cells pretreated with transforming growth factor-β (TGF-β). A more in-depth study revealed that miR-618 targeted and negatively regulated Smad4 to attenuate the progression of liver fibrosis [68]. BMSC-derived exosomes effectively alleviated liver fibrosis by inhibiting the Wnt/β-catenin signaling pathway to reduce collagen accumulation, suppress inflammation and enhance liver function and hepatocyte regeneration [69]. However, Chiabotto et al. reported a different result. MSC-derived EVs increased the expression of profibrotic markers in activated LX-2 cells compared to human hepatic stem cells, which may be due to the presence of COL1α1 in MSC-derived EVs [70]. MSC-derived exosomes may ameliorate liver fibrosis, similar to parental cells, but ambiguity remains. Therefore, the mechanism of amelioration must be elucidated by more in-depth studies (including characterization of the contents).

### 4.3. MSC-Derived Exosomes for the Treatment of Acute Liver Failure

Liver failure is clinically classified into four categories according to histopathological features and the speed of disease progression: acute, subacute, acute-nonchronic, and chronic liver failure [71]. Liver failure is characterized by an acute exacerbation of compensated or decompensated chronic liver disease, which is triggered by many factors, including viruses and hepatotoxic chemicals [72]. Acute liver failure (ALF) is a syndrome of multiple etiologies that is pathologically characterized by massive hepatocyte death, which results in severe impairment or decompensation of liver function and failure of other organs. These failures induce a series of clinical syndromes, such as coagulation disorders, hepatic encephalopathy and ascites, with high short-term mortality [73].

ALF is a process of hepatocyte injury that is dominated by inflammatory reactions, and a variety of hepatotoxic factors, such as Con-A, acetaminophen (APAP) and LPS, induce immune dysfunction and lead to ALF [74,75,76]. MSC-derived exosomes reduce the inflammatory response. Adipose tissue-derived MSC (AMSC)-derived exosomes and HUC-MSC-derived exosomes had positive effects in the treatment of D-GalN/LPS-induced ALF. Both exosomes reduced alanine aminotransferase (ALT) and aspartate aminotransferase (AST) levels in serum to varying degrees, and reduced the secretion of inflammatory factors (including IL-1β, IL-6, IL-18, TNF-α and TNF-γ) by inhibiting activation of the NOD-like receptor thermal protein domain associated protein 3 (NLRP3) inflammasome in macrophages to improve ALF [77,78]. HUC-MSC-derived exosomes also had therapeutic effects on APAP-induced ALF, which exerted hepatoprotective effects via anti-inflammatory and antioxidant aspects by downregulating the expression of inflammatory cytokines (such as IL-6, IL-1β, and TNF-α), upregulating extracellular regulated protein kinases (ERK)1/2 and phosphoinositide 3-kinase (PI3K)/AKT signaling pathways, and inhibiting oxidative stress-induced hepatocyte and LO2 cell apoptosis [79]. MSC-derived exosomes play an equally important role in subsequent angiogenesis and liver regeneration. C-reactive protein (CRP) in exosomes secreted by placenta-derived MSCs (PD-MSCs) acted as a trigger factor to activate the Wnt signaling pathway and upregulate von Willebrand Factor, vascular endothelial growth factor (VEGF), and vascular endothelial growth factor receptor 2 (VEGFR2) expression, which are involved in angiogenesis and liver regeneration [80]. In conclusion, MSC-derived exosomes exhibit therapeutic potential for ALF, which are primarily attributed to their anti-inflammatory and regulatory signal transduction abilities.

### 4.4. MSC-Derived Exosomes for the Treatment of Hepatocellular Carcinoma

Hepatocellular carcinoma (HCC) is a common and deadly cancer that accounts for approximately 90% of all liver cancer cases [81]. Although the continuous improvement of treatment methods, including surgical resection, orthotopic liver transplantation, transcatheter arterial chemoembolization, systemic or regional chemotherapy, and targeted immunotherapy, have greatly curbed the development of HCC, the postoperative recovery of patients is disappointing [82]. The development and metastasis mechanisms of HCC have been described. One mechanism is the communication between tumor cells and other cells during the formation of HCC, which promotes its growth and metastasis [83]. Exosomes are involved in intercellular communication, and perhaps a new therapeutic approach may be developed with the help of this function.

HCC is a primary malignancy of the liver. The tumor microenvironment (TME) assists tumor cells in establishing communication with neighboring and distant cells and plays an important role in the growth and metastasis of the primary tumor. The TME includes the extracellular matrix, endothelial cells, cancer-associated fibroblasts, immune cells, and MSCs [84]. Exosomes play multiple roles in the TME, and the mediation of communication is critical in influencing cancer cell survival. MSC-derived exosomes carry abundant proteins and RNAs, possess the tumor-regulating properties of parental cells, and play different regulatory roles in tumorigenesis, angiogenesis, invasion, migration, and drug resistance [85]. Secondary administration of ADMSC-derived exosomes to rats with HCC significantly reduced tumor volume and increased the mean percentage of circulating natural killer T (NKT) cells to facilitate HCC suppression, an early apparent diffusion coefficient (ADC) increase, and low-grade tumor differentiation [86]. Epithelial-to-mesenchymal transition (EMT) is the process by which epithelial cells with normal basement membrane interactions lose their epithelial characteristics and form a more migratory mesenchymal phenotype. EMT enhances cell migration, invasiveness and resistance to apoptosis and induces angiogenesis, which is associated with cancer development [87,88]. HUC-MSC-derived exosomal miR-451a may play a role in inhibiting the EMT process in HCC cells by targeting metalloprotease 10 (ADAM10) [89]. Two other studies also demonstrated that the BMSC-derived exosomes miR-15a and miR-338-3p delayed the proliferation, migration and invasion of cancer cells by downregulating spalt-like transcription factor 4 (SALL4) and E26 transformation specific-1 (EST1), respectively [90,91]. There is increasing evidence that MSC-derived exosomes are closely associated with the development, progression and amelioration of HCC. MSC-derived exosome-mediated immune regulation and HCC invasion and metastasis are essential for TME shaping and EMT [92]. Therefore, further elucidation of their physiological activities in the hepatic TME and potential mechanisms for EMT will help explore the MSC-derived exosome potential for improved treatment of HCC.

The goal of chemotherapy for liver cancer is to deliver drugs into the TME and kill cancer cells with minimal toxicity. However, most chemotherapy drugs also kill normal tissue cells, especially blood and lymphoid tissue cells, which is unfavorable for the body’s immune regulation. Therefore, it is particularly important for target drugs to kill cancer cells using suitable drug delivery vehicles for the treatment of liver cancer.

As biomedical research continues to intensify, many potential targets for the treatment of liver disease have been identified. Liposomes are a relatively well-established carrier for targeted drugs that facilitate the stabilization of therapeutic compounds to overcome barriers to cellular and tissue uptake and improve the biodistribution of compounds at target sites in vivo. However, liposomes have some limitations, including the toxicity of liposome membranes and the low biocompatibility of targeting ligands [93]. Notably, MSC-derived exosomes are a natural nanolipid bilayer composition that is less immunogenic than liposomes and has good tolerance and safety. These exosomes carry drugs across biological membranes and protect enzymes or RNA from degradation. Exosomes are taken up by target cells via endocytosis or membrane fusion and act on intracellular or external targets [21,25]. This process suggests that MSC-derived exosomes may act as special drug delivery vehicles for targeted drug delivery and enhance drug binding to receptor cells to maximize drug efficacy.

Therapeutic “cargos” (including small molecule drugs, DNA, RNA, proteins and probes for diagnostic therapy or imaging, specific targeting ligands and conjugation bonds integrated into exosomes) may be loaded into exosomes in three ways: (1) isolating donor cell-derived exosomes and integrating the therapeutic into the exosomes; (2) integrating the therapeutic into the donor cell and obtaining the therapeutic in the derived exosomes; and (3) transfecting or infecting the donor cell with DNA encoding the therapeutically active compound and releasing the therapeutic in the exosomes [94,95,96]. The choice of different integration modalities correlates with the type of therapeutic agent, the type of donor cells, and the pathological setting. The location of cargo loading into exosomes also varies. Hydrophobic and hydrophilic drugs are loaded in the lipid bilayer and aqueous phase, respectively. DNA, RNA, and proteins are contained in exosomes, and probes for diagnostic therapy or imaging, specific targeting ligands, and conjugate bonds are embedded on the surface of exosomes (Figure 3).

MSC-derived exosomes are increasingly used as drug delivery vehicles in tumor therapy and regenerative medicine [97]. Tumor cell drug resistance and drug insensitivity are difficult problems in treatment, and the use of MSC-derived exosomes as a drug delivery tool may provide new solutions to these problems. Glucose-regulated protein/BiP (GRP78) is overexpressed in cancer cells that are resistant to the anticancer drug sorafenib. SiGRP78-modified MSC-derived exosomes bound to sorafenib, targeted GRP78 in HCC cells and inhibited HCC cell growth, invasion and metastasis in vivo and in vitro, and reversed the resistance of cancer cells to sorafenib [98]. Improving the chemosensitivity of tumor cells to drugs is an equally urgent problem for tumor therapy. Lou et al. demonstrated that intratumor injection of miR-122-modified AMSC-derived exosomes significantly improved the antitumor efficacy of a chemotherapeutic agent (sorafenib) for HCC [99]. Further studies showed that AMSC-derived exosomes modified with miR-199a effectively enhanced the sensitivity of HCC cells to chemotherapeutic agents (doxorubicin) by targeting the mTOR signaling pathway [100]. Loading of the anticancer drug norethindrone into BMSC-derived exosomes using electroporation achieved a sustained slow release of the drug and effectively promoted HepG2 cell uptake, reduced tumor cell proliferation, and induced apoptosis. Notably, the drug-encapsulated exosomes repaired damaged liver tissue, which was manifested by increased hepatocyte proliferation and inhibition of oxidation [101]. In conclusion, MSC-derived exosomes as drug carriers in combination with anticancer drugs enhance drug efficacy, inhibit tumor progression and promote liver tissue repair.

## 5. Clinical Trials Using MSCs for the Treatment of Liver Diseases

Many clinical trials have been performed using MSCs for the treatment of liver disease and focused on clinical trial design, cell source, injection modality, and treatment efficacy [102,103,104]. Clinical studies related to the treatment of liver disease with MSCs were searched using https://clinicaltrials.gov/ (accessed on 18 June 2021) (Table 1). This type of clinical study admits patients of different ages with autologous or allogeneic transplanted MSCs for long-term treatment to evaluate the safety and efficacy of MSCs by detecting changes in the physiological and pathological parameters of patients at the end of treatment. For different types of liver disease, despite the different sources and injection routes of MSCs, the investigators examined their safety and efficacy by primarily testing the patients’ ALT and AST levels, the incidence of adverse effects and the survival time of patients after receiving treatment. The investigators also determined the effectiveness of MSC treatment by observing changes in the clinical symptoms of patients and assessing the Child–Pugh score and changes in liver function. Notably, the therapeutic effects of different sources of MSCs and different routes of injection for the treatment of liver disease were also studied. Zhang et al. admitted 210 patients with HBV-related liver failure and treated them with BM-MSCs and UC-MSCs. Patient survival, changes in liver function, degree of hepatic necrosis and improvement in symptoms were observed at the end of the treatment phase to analyze the safety and efficacy of transplantation of multiple MSCs for liver failure (NCT01844063). Shi et al. recruited 200 subjects with liver cirrhosis who were randomly divided into three groups. UC-MSCs were infused into patients using the interventional method via the hepatic artery for one group and intravenously for another group. The control group received conservative therapy. ALT, total bilirubin, prothrombin time, prealbumin, albumin, and overall survival were examined at the end of the phase to compare the efficacy of the different interventions (NCT01233102). These findings suggested that MSCs had a positive effect in the treatment of liver disease. Therefore, to more effectively exploit the potential of MSCs for the treatment of liver disease, there is a need to standardize the delivery route of MSCs, optimize the therapeutic dose of MSCs, elucidate the therapeutic mechanism of MSCs, and prolong the bioactive time of transplanted MSCs [1].

Unfortunately, there are few clinical investigations on MSC-derived exosomes for the treatment of liver diseases. However, based on the excellent performance of their parental cells in clinical treatment, we boldly speculate that MSC-derived exosomes are likely to exert promising therapeutic effects in the clinical setting [105]. Similarly, the problems to be solved for MSC-derived exosomes are similar to the problems for MSCs. Notably, MSC-derived exosomes have a unique advantage because they may be used as a low toxicity biomarker to enable more accurate tracking of transplants [106]. In conclusion, MSC-derived exosomes have great promise for the clinical treatment of liver diseases and exhibit similar or improved safety and efficacy than parental cells. The development of exosomes as a biomarker may further exploit their therapeutic potential.

## 6. MSC-Derived Exosomes: A Double-Edged Sword for Disease Treatment

The microenvironment around tumor cells includes the ECM, soluble factors, and various tumor stromal cells, such as MSCs, fibroblasts, and immune cells, which affect the occurrence and growth of tumors [107]. MSCs are one of the important components of tumor stromal cells, which may support the formation of the TME [108]. MSC-derived exosomes are involved in the communication between tumor and nontumor cells and have potential regulatory effects on tumor suppression and support.

Angiogenesis is one of the fundamental programs involved in tumorigenesis, and inhibiting angiogenesis in cancer cells has become a desirable approach. MSC-derived exosomes are rich in miR-16 (a miRNA targeting VEGF), which significantly downregulates VEGF expression in breast cancer cells by transferring antiangiogenic molecules, and results in the inhibition of angiogenesis and tumor progression [109]. MiR-100 is also enriched in MSC-derived exosomes, which mediate the downregulation of VEGF expression by regulating the mTOR/HIF-1α/VEGF signaling axis in breast cancer cells to inhibit angiogenesis in vitro [110]. MSC-derived exosomes also affect tumor progression by inhibiting the proliferation and promoting the apoptosis of cancer cells. MSC-derived exosomes carrying miR-133b inhibited glioma cell proliferation, invasion and migration, which are associated with enhancer of zeste 2 polycomb repressive complex 2 subunit (EZH2) silencing and inhibition of the Wnt/β-catenin signaling pathway [111]. Inhibition of rhophilin Rho GTPase binding protein 2 (RHN2) by HBMSC-derived exosomal miR-205 inhibited prostate proliferation and enhanced the apoptosis of cancer cells, which suggests that miR-205 is a predictor and potential therapy for prostate cancer targets [112]. The contents of exosomes are rich and diverse, and the specific mechanisms by which miRNAs and other cargoes inhibit tumor progression remain to be elucidated.

Although MSC-derived exosomes have shown great therapeutic potential in some basic studies, they are also lacking, especially in tumor therapy, due to their parental cells’ ability to produce a broad spectrum of cytokines, chemokines and growth factors, which are key cells that may promote tumor growth [113,114]. MSC-derived exosomes carry and deliver mRNAs, miRNAs and proteins that act directly on tumor cells to promote their growth and alter the phenotypic and functional characteristics of nontumor cells in the TME to enhance their tumor-promoting ability [115]. The promotion of tumor development by MSC-derived exosomes was confirmed in relevant basic studies (Table 2). BMSC-derived exosomes promoted tumor-associated cell growth by activating the Hedgehog signaling pathway [116]. Several microRNAs also had a positive impact on tumor development. MiR-208a is involved in the proliferation, migration and invasion of osteosarcoma cells via the downregulation of PDCD4 and activation of the ERK1/2 pathway [117]. MiR-19b-3p targets SOCSI to promote esophageal cancer progression [118]. MiR-193a-3p, miR-210-3p and miR-5100 activate the STAT3 pathway and induce EMT, and promote lung cancer cell invasion [119]. Gastric cancer tissue-derived mesenchymal stem cells can favor gastric cancer progression by transferring exosomal miRNAs to gastric cancer cells [120]. MSC-derived exosomes also enhance the expression of VEGF in tumor cells and promote their growth via activation of the ERK1/2 pathway [121]. Activation of the ERK1/2 pathway and overexpression of VEGF are clearly observed in liver cancer and liver fibrosis [122,123]. Because of the molecular mechanism exhibited by MSC-derived exosomes in other diseases, they may promote the occurrence and development of liver diseases. Basic studies have shown that MSC-derived exosomes exerted inhibitory and promotive effects on disease development. Therefore, the exact association of MSC-derived exosomes with diseases remains controversial, which directly affects clinical applications. These problems cannot be ignored, even as a drug delivery vehicle [124].

## 7. Current Challenges

The potential of MSC-derived exosomes for the treatment of liver disease and other classes of diseases is exciting, but the difficulties are challenging. Although MSC-derived exosomes are relatively easy to obtain, MSCs themselves have a limited number of in vitro passages, which results in limited access to exosomes. Regrettably, there is no apparent optimal isolation and purification technique to obtain a sufficient amount of highly pure exosomes [125]. Some researchers creatively combined microcarrier-based 3D cell culture technology with tangential flow filtration to develop a robust and scalable strategy to increase exosome yield 140-fold [126]. However, this yield was only obtained in the laboratory and may not be sufficient for future standardized production for clinical use. Future applications of MSC-derived exosomes are most likely in combination with other drugs or systems. Therefore, the therapeutic mechanisms of exosome intervention in disease onset must be further elucidated. The contents and levels of secreted exosomes are altered after MSCs are treated with serum starvation and hypoxia [127,128], which makes clinical treatment more difficult. Therefore, a comprehensive analysis of the chemical and functional characterization of exosomes and study of the physiological properties, diversity, and mode of transport of the contents are necessary. Notably, the development of exosomes as therapeutic delivery vehicles requires good manufacturing practices for medical product (GMP)-compatible production and purification methods and the development of new methods for the efficient loading of drugs into exosomes. A well-established GMP has a profound impact on establishing the protein profile, RNA profile, characterization analysis and biodistribution of exosomes [129,130,131]. These new methods are clearing the way for future clinical applications.

## 8. Conclusions

A large amount of research devoted to MSC-derived exosomes offers great promise and new ideas as a cell-free therapy and drug delivery vehicle for the treatment of liver disease and other diseases. However, there are challenges, including characterization of contents, specific molecular mechanisms of disease treatment and biosafety as a drug delivery system, that must be addressed to ensure their safety and efficacy. In conclusion, more basic research support and the combination of new technologies are needed to fully realize the therapeutic potential of MSC-derived exosomes and accelerate their application in the clinic.

## Figures and Tables

**Figure 1 ijms-23-10972-f001:**
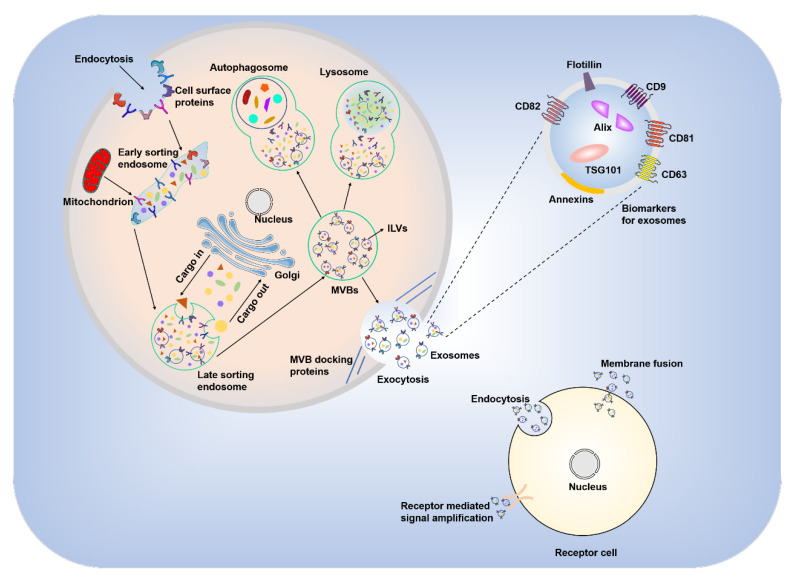
Exosomes biogenesis, surface markers and information exchange patterns. The cell membrane invaginates to form early endosomes, and multiple proteins, RNA, and DNA continue to accumulate to form late endosomes, further forming MVBs, which fuse with the plasma membrane to form and release exosomes. The biomarkers of exosomes include CD9, CD81, CD63, CD82, flotillin, annexins, Alix, TSG101, etc. Recipient cells will take up exosomes through cytokinesis, membrane fusion and receptor-mediated internalization. MVBs: Multivesicular bodies; ILVs: Intraluminal vesicles; TSG101: Tumor susceptibility gene 101 protein; Alix: ALG-2-interacting protein X.

**Figure 2 ijms-23-10972-f002:**
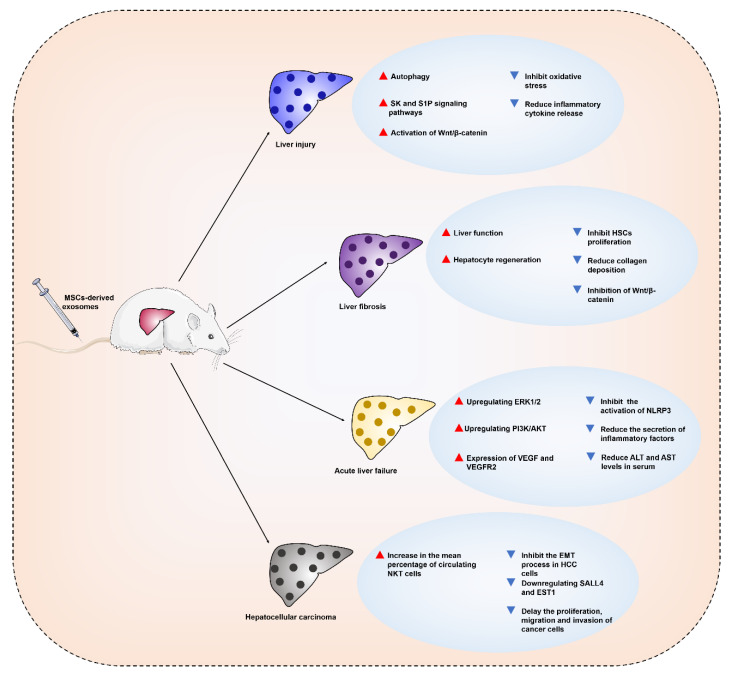
Pathological changes and molecular mechanisms involved in MSCs-derived exosomes in the treatment of liver diseases. Due to the different pathogenesis and pathological characteristics of various liver diseases, the main ways in which MSCs-derived exosomes exert their therapeutic effects are also different. SK: sphingosine kinase; S1P: sphingosine-1-phosphate; ERK1/2: extracellular regulated protein kinases1/2; PI3K/AKT: phosphoinositide 3-kinase/ protein kinase B; NLRP3: NOD-like receptor thermal protein domain associated protein 3; SALL4: spalt-like transcription factor 4; EST1: E26 transformation specific-1.

**Figure 3 ijms-23-10972-f003:**
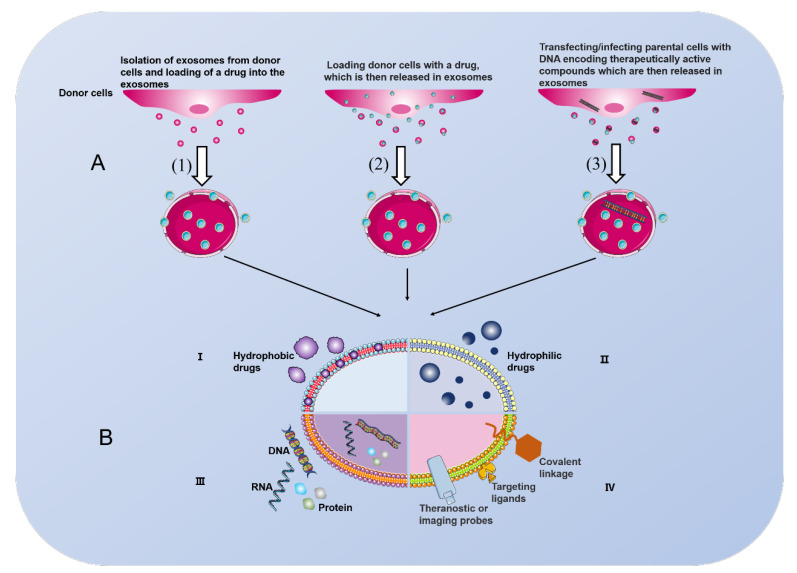
(**A**): Three different ways to load therapeutic “cargos” into exosomes. (1) Solation of exosomes from donor cells and integration of drugs with them. (2) Integration of drugs into donor cells, where the drugs are contained in their secreted exosomes. (3) Encoded DNA is transfected into donor cells and “cargo” is released in exosomes. (**B**): Different types of exosome drug delivery systems. Hydrophobic (I) or hydrophilic (II) compounds are bound in the lipid bilayer or in the aqueous phase, respectively. Exosomes can also be used for DNA, RNA, and protein delivery (III). Probes, specific targeting ligands and linker bonds embedded on the surface of exosomes (IV).

**Table 1 ijms-23-10972-t001:** Part of clinical studies related to MSCs for liver diseases.

Diseases	Source of MSCs	Injection Route	Test Evaluation Subjects	Phase	ClinicalTrials.gov Identifier
HBV-related acute-on-chronic liver failure	UCB-MSCs	Peripheral veins	Safety and efficacy	Phase II	NCT02812121
Liver failure caused by hepatitis B	Autologous MMSCs	The proper hepatic artery	The short-term efficacy and long-term prognosis	Unknown	NCT00956891
Acute-on-chronic liver failure caused by HBV	HUC-MSCs	Peripheral vein	Safety and efficacy	Phase I/II	NCT01724398
HBV-related liver failure	Allogeneic HBM-MSCs and HUC-MSCs	Peripheral vein	Safety and efficacy	Phase I/II	NCT01844063
The early and middle stage of liver cirrhosis on the basis of HBV infection	Autologous BMSCs	Portal vein	The therapeutic efficacy	Phase II	NCT00993941
HBV-related liver cirrhosis	Autologous BMSCs	Hepatic artery	Safety and efficacy	Phase I/II	NCT01724697
Liver cirrhosis resulting from chronic HBV	Allogenic BMSCs	Portal vein or hepatic artery	The therapeutic efficacy	Phase II	NCT01223664
Liver cirrhosis with refractory ascites	Autologous BMSCs	Liver artery	The effect of MSCs in the patients	Phase III	NCT01854125
Alcoholic liver cirrhosis	BMSCs	Hepatic artery catheterizations	Safety and efficacy	Phase II	NCT01741090
Liver cirrhosis	UC-MSCs	Intravenously	Safety and efficacy	Phase I/II	NCT01220492
Decompensated cirrhosis	HUC-MSCs	Intravenously	Safety and efficacy	Phase II	NCT05224960
Liver cirrhosis due to biliary atresia	UC-MSCs	Hepatic artery	Safety and efficacy	Phase I/II	NCT04522869
Liver cirrhosis	UC-MSCs	Hepatic artery	Safety and efficacy	Phase I/II	NCT01224327
Liver cirrhosis	UC-MSCs	Hepatic artery and intravenously	The efficacy of different interventional therapies	Phase I/II	NCT01233102

MMSCs: marrow mesenchymal stem cells; HUC-MSCs: human umbilical cord MSCs; UCB-MSCs: umbilical cord blood mesenchymal stem cells.

**Table 2 ijms-23-10972-t002:** Studies showing promotion of tumor progression by MSCs-derived exosomes.

Source of Exosomes	Study Purposes	Study Results	References
Human BMSC-derived exosomes	To study the molecular mechanism of MSCs on the growth of human osteosarcoma and human gastric cancer cells	Promoting MG63 and SGC7901 cell growth through the activation of Hedgehog signaling pathway	[116]
BMSC-derived exosomes	To explore the impact of the miR-208a-enriched exosomes derived from BMSCs on osteosarcoma cells.	Promoting proliferation, migration and invasion of osteosarcoma cells partly through downregulation of PDCD4 and activation of ERK1/2 pathway	[117]
Adult BMSC-derived exosomes	To explore the role of BMSC-derived exosomes mediating miR-19b-3p in EC cell growth	BMSC-derived exosomal microRNA-19b-3p targets SOCS1 to facilitate progression of esophageal cancer	[118]
Hypoxic BMSC-derived exosomes	To further elucidate the communication between BMSC-derived exosomes and cancer cells in the hypoxic niche	Exosome-mediated transfer of miR-193a-3p, miR-210-3p and miR-5100, could promote invasion of lung cancer cells by activating STAT3 signaling-induced EMT	[119]
Exosomes derived from gastric cancer tissue-derived MSCs	To explore the expression and role of miRNAs in gastric cancer stromal cells	Favouring gastric cancer progression by transferring exosomal miRNAs to gastric cancer cells	[120]
Human BMSC-derived exosomes	To explore the mechanism of MSCs xenograft promoting tumor growth	Enhancing VEGF expression in tumor cells by activating ERK1/2 pathway to promote tumor growth in vivo	[121]

BMSCs: bone mesenchymal stem cells; PDCD4: programmed cell death 4; miR: microRNA; SOCS1: suppressor of cytokine signaling 1; STAT3: signal transducer and activator of transcription 3; EMT: epithelial-to-mesenchymal transition; ERK1/2: extracellular regulated protein kinases1/2.

## Data Availability

Not applicable.

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
