# Peer review of "Mesenchymal Stem Cell-Derived Exosomes: A Promising Therapeutic Agent for the Treatment of Liver Diseases"

_ijms, 2022, doi:10.3390/ijms231810972_

Round 1

Reviewer 1 Report

Ms ID: ijms-1905303

Ms title: Mesenchymal stem cells-derived exosomes: a promising thera- 2 peutic agent for the treatment of liver diseases

Major comments:

This article is well written and scientific sound. However, there are some issues need revision before publication.

Minor comments:

1. Please including the reference citation in the caption of Figure 1.

2. “There is increasing evidence that MSCs-derived exosomes are closely associated with the development, progression and amelioration of HCC. MSCs- derived exosomes-mediated immune regulation and HCC invasion and metastasis are essential for TME shaping and EMT, therefore, further elucidation of their physiological activities in hepatic TME and potential mechanisms for EMT can help to explore the MSCs-  derived exosomes’ potential for improved treatment of HCC” (line 319-line 324). Please including reference citation in each report for this section

3. Please including the reference citation in the caption of Figure 2.

4. Please including the reference citation in the caption of Figure 3.

5. “For different types of liver disease, despite the different sources and injection routes of MSCs, the investigators examined their safety and efficacy, mainly testing the patients' ALT and AST levels, the incidence of adverse effects and the survival time of patients after receiving treatment. In addition, the investigators also determined the effectiveness of treatment of MSCs by observing the change in clinical symptoms of patients, assessing the Child-Pugh score, and changes in liver function. It is worth noting that the therapeutic effects of different sources of MSCs and different routes of injection for the treatment of liver disease have also been studied by researchers. Zhang's research group admitted 210 patients with HBV-related liver failure and treated them with BM-MSCs and UC-MSCs, respectively. Patient survival, changes in liver function, degree of hepatic necrosis and improvement in symptoms were observed at the end of the treatment phase to analyze the safety and efficacy of transplantation of multiple MSCs for liver failure (NCT01844063). Shi's research team recruited 200 subjects with liver cirrhosis who were randomly divided into 409 three groups. UC-MSCs were infused into patients using interventional method via hepatic artery for one group and were infused into patients intravenously for another group. The control group would receive conserved therapy. ALT, total bilirubin, pro-thrombin time, prealbumin, albumin, and Overall survival were examined at the end of the phase to compare the efficacy of the different interventions (NCT01233102). These findings suggest that MSCs have a positive effect on the treatment of liver disease.” (Line 396-415). Please including reference citation in each report for this section. 

Author Response

Response to Reviewer 1 Comments

Point 1:  Please including the reference citation in the caption of Figure 1.

Response 1: This figure is designed and drawn independently by us summarizing many literatures. The content involved in the figure has been reflected in the article.

Point 2: “There is increasing evidence that MSCs-derived exosomes are closely associated with the development, progression and amelioration of HCC. MSCs- derived exosomes-mediated immune regulation and HCC invasion and metastasis are essential for TME shaping and EMT, therefore, further elucidation of their physiological activities in hepatic TME and potential mechanisms for EMT can help to explore the MSCs- derived exosomes’ potential for improved treatment of HCC” (line 319-line 324). Please including reference citation in each report for this section

Response 2: We have added corresponding reference citations. details as follows,

 Liu, J. et al. The biology, function, and applications of exosomes in cancer. Acta pharmaceutica Sinica. B. 11(9), 2783–2797 (2021).

Point 3:  Please including the reference citation in the caption of Figure 2.

Response 3: This figure is designed and drawn independently by us summarizing many literatures. The content involved in the figure has been reflected in the article.

Point 4:  Please including the reference citation in the caption of Figure 3.

Response 4: This figure is designed and drawn independently by us summarizing many literatures. The content involved in the figure has been reflected in the article.

Point 5: “For different types of liver disease, despite the different sources and injection routes of MSCs, the investigators examined their safety and efficacy, mainly testing the patients' ALT and AST levels, the incidence of adverse effects and the survival time of patients after receiving treatment. In addition, the investigators also determined the effectiveness of treatment of MSCs by observing the change in clinical symptoms of patients, assessing the Child-Pugh score, and changes in liver function. It is worth noting that the therapeutic effects of different sources of MSCs and different routes of injection for the treatment of liver disease have also been studied by researchers. Zhang's research group admitted 210 patients with HBV-related liver failure and treated them with BM-MSCs and UC-MSCs, respectively. Patient survival, changes in liver function, degree of hepatic necrosis and improvement in symptoms were observed at the end of the treatment phase to analyze the safety and efficacy of transplantation of multiple MSCs for liver failure (NCT01844063). Shi's research team recruited 200 subjects with liver cirrhosis who were randomly divided into 409 three groups. UC-MSCs were infused into patients using interventional method via hepatic artery for one group and were infused into patients intravenously for another group. The control group would receive conserved therapy. ALT, total bilirubin, pro-thrombin time, prealbumin, albumin, and Overall survival were examined at the end of the phase to compare the efficacy of the different interventions (NCT01233102). These findings suggest that MSCs have a positive effect on the treatment of liver disease.” (Line 396-415). Please including reference citation in each report for this section. 

Response 5: We are happy to accept your valuable suggestions. Each clinical report in this section is sourced from clinicaltrials.gov, and we provide NCT numbers. Therefore, we believe this is sufficient for readers looking for raw data.

Sincerely thank you for your advice again.

Reviewer 2 Report

The article by Ding et al. entitled "Mesenchymal stem cells-derived exosomes: a promising therapeutic agent for the treatment of liver diseases" is a review of the current state of knowledge on the possible use of mesenchymal stem cell-derived exosomes for the treatment of liver diseases. Although similar reviews exist in the literature, the update on this topic is interesting because of the continuous progress in research on the use of MSC-derived exosomes in the treatment of different pathologies, including those affecting the liver.

Remarks

11.     Lines 103-104. The text says: "Exosomes contain a variety of molecules, including proteins, lipids, DNA, RNA, and microRNA (miR-),....". When the authors say RNA, this includes microRNA. Therefore I suggest that if they want to emphasize the content of the presence of microRNA, they should express it in another way. For example: "Exosomes contain a variety of molecules, including proteins, lipids, DNA and RNA. Among the different types of RNA, the presence of microRNAs stands out...".

22.       The section "4.4. MSCs-derived exosomes for treating hepatocellular carcinoma" describes the studies on the treatment with exosomes of the main cancer affecting the liver. In the following section: "5. MSCs-derived exosomes as drug delivery vehicles for treating liver cancer", after a general description of the potential of exosomes to transport drugs, the information focuses again on the use of exosomes for treating hepatocellular carcinoma. In my opinion it would be convenient to combine the two sections.

33.       Section 7 (MSCs-derived exosomes: a double-edged sword for treating diseases) is interesting because it shows the risk of using exosomes for therapeutic purposes due to their possible capacity to promote tumor cell progression. However, it would be interesting that the authors had also provided information on studies showing that MSCs-derived exosomes may also have antitumor capacity or that they can be manipulated to treat certain cancers (see for example ref. DOI 10.1186/s13045-021-01141-y). I believe that this would give more information to the reader on the importance of further research on the mechanism of action of exosomes and their cargo, so that they can be safely used in human clinical practice.

44.       In Table 2, I suggest that the title may be changed to another one such as: "Studies showing promotion of tumor progression by MSCs-derived exosomes".

Author Response

Response to Reviewer 2 Comments

Point 1:   Lines 103-104. The text says: "Exosomes contain a variety of molecules, including proteins, lipids, DNA, RNA, and microRNA (miR-),....". When the authors say RNA, this includes microRNA. Therefore I suggest that if they want to emphasize the content of the presence of microRNA, they should express it in another way. For example: "Exosomes contain a variety of molecules, including proteins, lipids, DNA and RNA. Among the different types of RNA, the presence of microRNAs stands out...".

Response 1: We accept the reviewers' valuable suggestions and make changes accordingly. Details as follows,

Exosomes contain a variety of molecules, including proteins, lipids, DNA and RNA. Among the different types of RNA, the presence of microRNAs (miR-) is notable because miRNAs are involved in cell adhesion, membrane fusion, metabolism, and signaling 28,29,30,31,32

Point 2: The section "4.4. MSCs-derived exosomes for treating hepatocellular carcinoma" describes the studies on the treatment with exosomes of the main cancer affecting the liver. In the following section: "5. MSCs-derived exosomes as drug delivery vehicles for treating liver cancer", after a general description of the potential of exosomes to transport drugs, the information focuses again on the use of exosomes for treating hepatocellular carcinoma. In my opinion it would be convenient to combine the two sections.

Response 2: We accept the reviewer's valuable suggestion to combine Sections 4.4 and 5. We use the following paragraphs as a transition between the two.

The goal of chemotherapy for liver cancer is to deliver drugs into the TME and kill cancer cells with minimal toxicity. However, most chemotherapy drugs also kill normal tissue cells, especially blood and lymphoid tissue cells, which is unfavorable for the body's immune regulation. Therefore, it is particularly important for target drugs to kill cancer cells using suitable drug delivery vehicles for the treatment of liver cancer.

Point 3: Section 7 (MSCs-derived exosomes: a double-edged sword for treating diseases) is interesting because it shows the risk of using exosomes for therapeutic purposes due to their possible capacity to promote tumor cell progression. However, it would be interesting that the authors had also provided information on studies showing that MSCs-derived exosomes may also have antitumor capacity or that they can be manipulated to treat certain cancers (see for example ref. DOI 10.1186/s13045-021-01141-y). I believe that this would give more information to the reader on the importance of further research on the mechanism of action of exosomes and their cargo, so that they can be safely used in human clinical practice.

Response 3: We gladly accept valuable suggestions from reviewers and have made revisions in the manuscript. Details as follows,

The microenvironment around tumor cells includes the ECM, soluble factors, and various tumor stromal cells, such as MSCs, fibroblasts, and immune cells, which affect the occurrence and growth of tumors 110. MSCs are one of the important components of tumor stromal cells, which may support the formation of the TME 111. MSC-derived exosomes are involved in the communication between tumor and nontumor cells and have potential regulatory effects on tumor suppression and support.

Angiogenesis is one of the fundamental programs involved in tumorigenesis, and inhibiting angiogenesis in cancer cells has become a desirable approach. MSC-derived exosomes are rich in miR-16 (a miRNA targeting VEGF), which significantly downregulates VEGF expression in breast cancer cells by transferring antiangiogenic molecules, and results in the inhibition of angiogenesis and tumor progression 112. MiR-100 is also enriched in MSC-derived exosomes, which mediated the downregulation of VEGF expression by regulating the mTOR/HIF-1α/VEGF signaling axis in breast cancer cells to inhibit angiogenesis in vitro 113. MSC-derived exosomes also affect tumor progression by inhibiting the proliferation and promoting the apoptosis of cancer cells. MSC-derived exosomes carrying miR-133b inhibited glioma cell proliferation, invasion and migration, which are associated with enhancer of zeste 2 polycomb repressive complex 2 subunit (EZH2) silencing and inhibition of the Wnt/β-catenin signaling pathway 114. Inhibition of rhophilin Rho GTPase binding protein 2 (RHN2) by HBMSC-derived exosomal miR-205 inhibited prostate proliferation and enhanced the apoptosis of cancer cells, which suggests that miR-205 is a predictor and potential therapy for prostate cancer targets 115. The contents of exosomes are rich and diverse, and the specific mechanisms by which miRNAs and other cargoes inhibit tumor progression remain to be elucidated.

Point 4:  In Table 2, I suggest that the title may be changed to another one such as: "Studies showing promotion of tumor progression by MSCs-derived exosomes".

Response 4: We accept the reviewers' valuable suggestions and make changes in the manuscript.

Sincerely thank you for your advice again.